# Are there researcher allegiance effects in diagnostic validation studies of the PHQ-9? A systematic review and meta-analysis

Laura Manea,[1,2] Jan Rasmus Boehnke,[3] Simon Gilbody,[1,2] Andrew S Moriarty,[2] Dean McMillan[1,2]

## ABSTRACT

**Objectives** To investigate whether an authorship effect is found that leads to better performance in studies conducted by the original developers of the Patient Health Questionnaire (PHQ-9) (allegiant studies).

**Design** Systematic review with random effects bivariate diagnostic meta-analysis. Search strategies included electronic databases, examination of reference lists and forward citation searches.

**Inclusion criteria** Included studies provided sufficient data to calculate the diagnostic accuracy of the PHQ-9 against a gold standard diagnosis of major depression using the algorithm or the summed item scoring method at cut-off point 10.

**Data extraction** Descriptive information, methodological quality criteria and 2×2 contingency tables.

**Results** Seven allegiant and 20 independent studies reported the diagnostic performance of the PHQ-9 using the algorithm scoring method. Pooled diagnostic OR (DOR) for the allegiant group was 64.40, and 15.05 for non-allegiant studies group. The allegiance status was a significant predictor of DOR variation (p<0.0001). Five allegiant studies and 26 non-allegiant studies reported the performance of the PHQ-9 at recommended cut-off point of 10. Pooled DOR for the allegiant group was 49.31, and 24.96 for the non-allegiant studies. The allegiance status was a significant predictor of DOR variation (p=0.015). Some potential alternative explanations for the observed authorship effect including differences in study characteristics and quality were found, although it is not clear how some of them account for the observed differences.

**Conclusions** Allegiant studies reported better performance of the PHQ-9. Allegiance status was predictive of variation in the DOR. Based on the observed differences between independent and non-independent studies, we were unable to conclude or exclude that allegiance effects are present in studies examining the diagnostic performance of the PHQ-9. This study highlights the need for future meta-analyses of diagnostic validation studies of psychological measures to evaluate the impact of researcher allegiance in the primary studies.

[1]Deparment of Health Sciences, University of York, York, UK
[2]Hull York Medical School, University of York, York, United Kingdom
[3]Dundee Centre for Health And Related Research, University of Dundee, Dundee, United Kingdom

**Correspondence to**
Dr Laura Manea;
laura.manea@york.ac.uk

## Strengths and limitations of this study

► An original study—the first meta-analysis of diagnostic validation studies of psychological measures to evaluate the impact of researcher allegiance.
► Using rigorous methodology—strict inclusion/exclusion and quality assessment criteria.
► We found that the allegiance effect was a significant predictor of the variation of the diagnostic OR in the meta-regression analysis.
► Substantial variability observed in methodological quality of included studies.
► Based on the observed methodological differences between the independent and non-independent studies, we were unable to conclude or exclude that allegiance effects are present in studies examining the diagnostic performance of the Patient Health Questionnaire (PHQ-9).

context, *allegiance* describes the phenomenon that researchers and clinicians who developed a treatment approach or are for other reasons invested in it tend to find larger effect sizes in favour of their treatment than for comparison groups.[1] This finding has been extensively replicated[2 3] and is also robust when the quality of research is controlled for. Researcher allegiance is subject of ongoing debates about the design of efficacy studies as well as implications for policy.[2 4 5] Researcher allegiance is also discussed widely in the literature on experimental as well as evaluation research.[6] Since the motivational underpinnings of allegiance effects are potentially far more ingrained into human behaviour and decision making than previously thought,[7] they may occur commonly in clinical research in general.

Although it has been suggested that allegiance effects may play a role in the validation of psychological screening and case-finding tools (eg, O'Shea *et al.*, in press), systematic

Research on allegiance effects has a long tradition in psychotherapy research. In this

evaluations of this hypothesis are rare and studies that acknowledge potential allegiance effects in such studies mainly come from forensic psychology and psychiatry backgrounds.[8–11] Diagnostic validation studies are geared at establishing the sensitivity and specificity of a screening or case-finding tool, which is used in practice to differentiate cases from non-cases or to decide about whether further assessment or treatment is indicated or will be offered. An allegiance effect in such studies would be seen in systematically higher sensitivities or specificities if the original author(s) is (are) part of the team of such a study. Such a bias would have a deleterious affect on practice through promising overoptimistic accuracy of the screening or case-finding tool or in evaluating the cost-effectiveness of the measure in a screening or case-finding context.

The depression module of the Patient Health Questionnaire (PHQ-9) is a widely used depression-screening instrument in non-psychiatric settings. The PHQ-9 was developed by a team of researchers, with its development underwritten by an educational grant from Pfizer US Pharmaceuticals.[12] The PHQ-9 can be scored using different methods, including an algorithm based on Diagnostic and Statistical Manual of Mental Disorders (DSM)-IV criteria and a cut-off based on summed-item scores. The psychometric properties of these two approaches have been summarised in two recently published meta-analyses.[13 14] The goal of the current review is to investigate, based on an established database of PHQ-9 diagnostic validation studies,[13 14] whether an allegiance effect is found that leads to an increased sensitivity and specificity in studies that were conducted by researchers closely connected to the original developers of the instrument.

## METHODS
### Study selection
Similar search strategies were used in both systematic reviews (for full details, please see Manea *et al* and Moriarty *et al*[13 14]). Embase, Medline and PsycINFO were searched from 1999 (when the PHQ-9 was first developed) to August 2013 and September 2013, respectively, using the terms 'PHQ-9', 'PHQ', 'PHQ$' and 'patient health questionnaire'. The search strategy is presented in online supplementary appendix 1. The reference lists of studies fitting the inclusion criteria were manually searched and a reverse citation search in Web of Science was performed. The authors of unpublished studies were contacted and conference abstracts were reviewed in an attempt to minimise publication bias.

The following inclusion-exclusion criteria were used:

*Population:* adult population. *Instrument:* studies that used the PHQ-9. *Comparison (reference standard):* the accuracy of the PHQ-9 had to be assessed against a recognised gold-standard instrument for the diagnosis of either DSM or International Classification of Disease (ICD) criteria for major depression. Studies were included if the diagnoses were made using a standardised diagnostic structured interview schedule (eg, Mini International Neuropsychiatric Interview (MINI), Structured Clinical Interview for DSM Disorders (SCID)). Unguided clinician diagnoses with no reference to a standard structured diagnostic schedule or comparisons of the PHQ-9 with other self-report measures were excluded. Studies were also excluded if the target diagnosis was not major depressive disorder (MDD, eg, any depressive disorder). *Outcome:* studies had to report sufficient information to calculate a 2×2 contingency table for the algorithm or the recommended cut-off point 10. *Study design:* any design. *Additional criterion:* we avoided double counting of evidence by ensuring that only one study of those that reported overlapping datasets in different journals were included in the meta-analysis. Citations with overlapping samples were examined to establish whether they contained information relevant to the research question that was not contained in the included report.

### Quality assessment
Quality assessment was performed using the Quality Assessment of Diagnostic Accuracy Studies (Revised) (QUADAS-2) tool, a tool for evaluating the risk of bias and applicability of primary diagnostic accuracy studies when conducting diagnostic systematic reviews.[15] It covers the areas of patient selection, index test, reference standard and flow and timing.[16] This tool was adapted for the two reviews and quality assessments were carried out by two independent reviewers for all studies included in the reviews.

### Data synthesis and statistical analysis
We constructed 2×2 tables for cut-off point 10[14] and the algorithm scoring method.[13] Pooled estimates of sensitivity, specificity, positive/negative likelihood ratios and diagnostic ORs (DOR) were calculated using random effects bivariate meta-analysis.[17] Heterogeneity was assessed using $I^2$ for the DOR, an estimate of the proportion of study variability that is due to between-study variability rather than sampling error. We considered values of ≥50% to indicate substantial heterogeneity.[18] Summary receiver operating characteristic curves (sROC) were constructed using the bivariate model to produce a 95% confidence ellipse within ROC space.[19] Each data point in the sROC space represents a separate study, unlike a traditional ROC plot, which explores the effect varying thresholds on sensitivity and specificity in a single study.

We undertook a meta-regression analysis of logit DOR using research allegiance as covariate in the meta-regression model.[20 21] Analyses were conducted using STATA V.12, with the metan, metandi and metareg user-written commands.

### Allegiance rating
We rated authorship on a paper if any of the developers of the PHQ-9—Kurt Kroenke, MD, Robert L Spitzer, MD and Janet BW Williams—as an indicator of potential allegiance. We also rated as evidence of allegiance as

acknowledged collaborations with the developers of the PHQ-9, even if they were not listed as coauthors or if the authors acknowledged funding from Pfizer to conduct the study.

## RESULTS
### Overview of included studies
Thirty-one studies reported the diagnostic properties of the PHQ-9 at cut-off point 10 or above and were included in this analysis.[14] Twenty-seven studies were included in the algorithm review.[13] The study selection flow charts can be found in online supplementary appendix 2 (figures 1 and 2). The characteristics of these studies are reported in tables 1 and 2 and the results of the methodological assessment are presented in tables 3 and 4.

### Algorithm scoring method
#### Descriptive characteristics
The descriptive characteristics of the included studies are presented in table 1. Seven individual studies that reported the diagnostic performance of the PHQ-9 using the algorithm scoring method were coauthored by the original developers of the PHQ-9,[22 26] specifically acknowledged one of the developers and support by an educational grant from Pfizer USA,[27] or were coauthored by the first author of a previous study that had also been coauthored by one of the developers.[28] Twenty non-allegiant studies reported the diagnostic properties of the PHQ-9 using the algorithm scoring method.

Three (43%, 3/7) of the allegiant studies were conducted exclusively in hospital settings.[22 26 28] The remaining four studies (67%, 4/7) were conducted in different settings or non-exclusively hospital settings: one in primary care[25] and three in mixed settings: psychosomatic walk in clinics and family practices,[23],[i] outpatient clinics and family practices[24] and primary care and hospital settings.[27] In the non-allegiant group, 13 (65%, 13/20) studies were conducted in hospital settings.[29–41] Of the remaining seven studies, six were conducted in primary care settings[42–47] and one in a community sample.[48]

In both groups (non-allegiant and allegiant studies), the majority of studies validated a translated version of the PHQ-9. Two of the studies authored by developers (28%, 2/7),[25 26] and eight (40%, 8/20) allegiant studies[29 30 37–40 42 48] were conducted in English.

The mean prevalence of MDD in the group of allegiant studies was 13.4% (range 6.1%–29.2%); in the non-allegiant group it was 15.5% (range 3.9%–32.4%). The mean age of patients in the PHQ-9 developers group was 45.7; all but one study had a mean age in the range of 40–50 years. In the non-allegiant group, the mean age was 54.6 (range 29.3–75.0), with almost half (8) of the studies reporting a mean age of over 60. The percentage of females in the PHQ-9 developers was 56.8% (range 28.6%–67.8%) and in the non-allegiant group was 59.1 (18%–100%).

All allegiant studies used a self-reported PHQ-9, whereas in seven non-allegiant studies (30%, 6/20) the PHQ-9 was administered by a researcher.[30–33 43 48] Apart from Muramatsu *et al.*, all allegiant studies used the SCID as a gold standard[27]; the non-allegiant studies used a wider range of gold standards including SCAN, CIDI, MINI and C-DIS, although the SCID was also frequently used by the independent studies as well (45%, 9/20 studies).

Four out of the seven allegiant studies (57%) did not include a conflict of interest statement.[22 23 25 27] Also, four (57%) of the allegiant studies acknowledged funding from Pfizer.[23–25 27] Only one study[27] acknowledged the collaboration with one of the developers of the PHQ-9.

Of the non-allegiant studies, 12 (60%) did not include a conflict of interest statement.[29–32 35–37 39 44–46 48] It appears that newer studies were more likely to include a conflict of interest statement, which may reflect a recent change in reporting. Funding was acknowledged by most studies (18/20) and most received funding from academic or/ and health research institutions. Two studies received funding from pharmaceutical companies—Lundbeck[43] and Pfizer[35] and one study acknowledged that Pfizer Italia provided the Italian version of PHQ-9 and gave the authors permission to use it.[36]

#### Diagnostic test accuracy
Pooled sensitivity and specificity was calculated separately for the non-allegiant and allegiant studies. Pooled sensitivity for the allegiant studies of the PHQ-9 was 0.77 (95% CI 0.70 to 0.84), pooled specificity was 0.94 (95% CI 0.90 to 0.97) and the pooled DOR was 64.40 (95% CI 34.15 to 121.43). Heterogeneity was high ($I^2$=78.9%). Figure 1 represents the sROCs for this set of studies.

Pooled sensitivity for the non-allegiant studies was lower compared with the developer authored studies group at 0.48 (95% CI 0.41 to 0.91), pooled specificity was the same at 0.94 (95% CI 0.91 to 0.95). The pooled DOR was approximately four times lower at 15.05 (95% CI 11.03 to 20.52) (see figure 1). Heterogeneity was substantial at $I^2$=68.1%.

The meta-regression analysis for algorithm studies with non-allegiant status as the predictor of the DOR showed that non-allegiant status was a significant predictor of the DOR ($p<0.0001$) and explained a substantial amount of the observed heterogeneity (51.5%).

#### Quality assessment
The results of the quality assessment using QUADAS-2 are given in table 3 for the studies reporting on the diagnostic performance of the algorithm scoring method. In the patient selection domain, more non-allegiant studies (65%, 13/20) than allegiant (29%, 2/7) met the criterion for consecutive referrals. There were no marked differences on the other two criteria in this domain (avoid

---

[i] This study provided separate estimates for the two settings in which it was conducted; therefore separate psychometric estimates were generated for each sample for both algorithm scoring method and summed items scoring method at cut-off point 10 (see below).

**Table 1** Descriptive characteristics of algorithm studies[13]

| Study | Sample characteristics | Sample size and % depressed | PHQ-9 characteristics | Diagnostic standard | a) COI declaration b) Funding c) Relationship with original developers |
|---|---|---|---|---|---|
| | (country, setting, age, sex) | | | | |
| Diez-Quevedo et al[22] | Country: Spain Setting: medical and surgical tertiary hospitals Age (years): M=43 (SD=14.2) Female: 45.6% | n=1003 Depressed: 8.2% | Administration: self-report Language: Spanish | DSM-III-R SCID | a) No COI declaration b) Funding acknowledged (academic institutions) c) Not acknowledged |
| Gräfe et al[23] | Country: Germany Setting: psychosomatic walk-in clinics and family practices Age (years): male=41.9 (SD=13.8) Female: 67.8% | n=528 Depressed: 29.2% psychosomatic patients; 6.16% medical patients | Language: German Administration: self-report | DSM-IV SCID | a) No COI declaration b) Acknowledged funding from Pfizer c) Not acknowledged |
| Lowe et al[24] | Country: Germany Setting: outpatient clinics and family practices Age (years): male=41.7 (SD=13.8) Female: 67.1% | n=501 Depressed: 13.2% | Administration: self-report Language: German | DSM-IV SCID | a) COI declaration 'This study was supported by unrestricted restricted grants from Pfizer Germany and from the medical faculty of the University of Heidelberg Germany, and there are no COI'. b) Acknowledged funding from Pfizer and academic institution c) Not acknowledged |
| Muramatsu et al[27] | Country: Japan Setting: primary care and general hospital Age (years): male=43.3 (SD=16.4) Female: 59.5% | n=131 Depressed: 28.2% | Administration: self-report Language: Japanese | DSM-IV MINI | a) No COI declaration b) Acknowledged funding from Pfizer c) Acknowledged one of the developers of the PHQ-9: 'The authors acknowledge Dr RL Spitzer' |
| Navinés et al[28] | Country: Spain Setting: general hospital (patients with chronic HCV) Age (years): male=43.4 (SD=10.2) Female: 28.6% | n=500 Depressed: 6.4% | Administration: self-report Language: Spanish | DSM-IV SCID | a) All authors declared that they had no COI. b) Role of funding source declared c) Not acknowledged |
| Spitzer et al[25] | Country: USA Setting: primary care Age (years): male=46 (SD=17.2) Female: 66% | n=3000 (585 received SCID) Depressed: 10% | Administration: self-report Language: English | DSM-III-R SCID | a) No COI declaration b) Acknowledged funding from Pfizer. 'Drs Spitzer and Williams receive honoraria and consulting money from Pfizer, which has supported this work'. c) N/A |
| Thekkumpurath et al[26] | Country: UK Setting: hospital (cancer patients) Age (years): male=61 Female: 63% | n=782 Depressed: 6.3% (of the whole sample) | Administration: not stated Language: English | DSM-IV SCID | a) COI declaration: 'Supported by Cancer Research UK' b) As in a) c) Not acknowledged |
| Ayalon et al[43] | Country: Israel Age (years): male=75 (SD=8.1) Female: 40.5% | n=153 Depressed: 3.9% | Administration: researcher administered Language: Hebrew | DSM-IV SCID | a) COI declaration: 'The project was funded by an Investigator's Initiated Research Grant from Lundbeck International given to Dr Liat Ayalon. Lundbeck International had no other involvement in the project concept of design or in this paper. Per Bech has occasionally over the past 3 years until August 2008 received funding from and has been speaker or member of advisory boards for pharmaceutical companies with an interest in the drug treatment of affective disorders (AstraZeneca, Lilly, H Lundbeck A/S, Lundbeck Foundation and Organon)'. b) Acknowledged funding from Lundbeck International |

**Table 1** Continued

| Study | Sample characteristics (country, setting, age, sex) | Sample size and % depressed | PHQ-9 characteristics | Diagnostic standard | a) COI declaration b) Funding c) Relationship with original developers |
|---|---|---|---|---|---|
| Eack et al[29] | Country: USA Setting: community mental health centres for children Age (years): male=39.20 (SD 9.63) Female: 100% | n=50 Depressed: 28% | Administration: self-report Language: English | DSM-IV SCID | a) No COI declaration b) Funding acknowledged (academic/health research institutions) |
| Fann et al[30] | Country: USA Setting: trauma hospital (inpatients with traumatic brain injury) Age (years): male=42 (SD=17.9) Female: 29.1% | n=135 Depressed: 16.3% | Administration: telephone-administered Language: English | DSM-IV SCID | a) No COI declaration b) Funding acknowledged (academic institutions) |
| Gelaye et al[31] | Country: Ethiopia Setting: general hospital Age (years): 34.9 (SD=11.6) Female: 63.1% | n=363 Depressed: 12.6% | Administration: researcher-administered Language: Amharic | DSM-IV SCAN | a) No COI declaration b) Funding acknowledged (academic/health research institutions) |
| Gjerdingen et al[48] | Country: USA Setting: community Age (years): male=29.3 Female: 100% | n=438 Depressed: 4.6% | Administration: telephone or self-report Language: English | DSM-IV SCID | a) No COI declaration b) Funding acknowledged (academic/health research institutions) |
| Henkel et al[44] | Country: Germany Setting: primary care Age (years): not reported Female: 74% | n=448 Depressed: 10% | Administration: self-report Language: German | DSM-IV CIDI | a) No COI declaration b) Funding acknowledged (academic/health research institutions) |
| Hyphantis et al[32] | Country: Greece Setting: hospital – rheumatology patients Age (years): male=54.2 (SD=13.5) Female: 74% | n=213 Depressed: 32.4% | Administration: researcher administered Language: Greek | DSM-IV MINI | a) No COI declaration b) No funding acknowledgement |
| Inagaki et al[33] | Country: Japan Setting: general hospital Age whole sample (years): male=73.5 (SD=12.3) Female: 59.3% | n=104 out of 511 received MINI Depressed: 7.4% | Administration: researcher administered Language: Japanese | DSM-IV MINI | a) COI declaration: 'The authors declare that they have no competing interests'. b) Funding acknowledged (academic/health research institutions) |
| Khamseh et al[34] | Country: Iran Setting: diabetes clinic Age (years): male=56.17 (SD=9.60) Female: 51.9% | n=185 Depressed: 43.2% | Administration:self report Language: Persian | DSM-IV SCID | a) COI declaration: the authors declared no competing interests b) Funding acknowledged (academic/health research institutions) |
| Lamers et al[45] | Country: The Netherlands Setting: primary care (elderly) Age (years): male=71.4 (SD=6.90) Female: 48.2% | n=713 Depressed: 10.7% | Administration:self report Language: Dutch | DSM-IV MINI | a) No COI declaration b) Funding acknowledged (academic/health research institutions) |

**Table 1** Continued

| Study | Sample characteristics (country, setting, age, sex) | Sample size and % depressed | PHQ-9 characteristics | Diagnostic standard | a) COI declaration b) Funding c) Relationship with original developers |
|---|---|---|---|---|---|
| Lotrakul et al[46] | Country: Thailand Setting: primary care Age (years): male=45.0 (SD=14.30) Female: 73.7% | n=279 Depressed: 6.8% | Administration:self report Language: Thai | DSM-IV MINI | a) No COI declaration b) Funding acknowledged (academic/health research institutions) |
| Persoons et al[35] | Country: Belgium Setting: hospital (otolaryngology patients) Age (years): male=48.2 (SD=12.9) Female: 65.6% | n=268 (97 received MINI) Depressed: 16.5% | Administration: self-report Language: Dutch | DSM-IV MINI | a) No COI declaration b) Funding acknowledged (academic/health research institutions and Pfizer Belgium |
| Picardi et al[36] | Country: Italy Setting: hospital (dermatology inpatients) Age (years): male=37.5 Female: 56% | n=141 Depressed: 8.5% | Administration: self-report Language: Italian | DSM-IV SCID | a) No COI declaration b) Funding acknowledged (academic/health research institutions) Acknowledged Pfizer Italia SRL for providing the Italian version of the PHQ-9 and for permission to use it. |
| Stafford et al[37] | Country: Australia Setting: hospital (cardiology patients) Age (years): male=64.1 (SD=10.3) Female: 66% | n=193 Depressed: 18% | Administration: self-report Language: English | DSM-IV MINI | a) No COI declaration b) Funding acknowledged (academic/health research institutions) |
| Thombs et al[38] | Country: USA Setting: hospital (outpatients with coronary heart disease) Age (years): male=67 (SD=11) Female: 18% | n=1024 Depressed: 22% | Administration: not stated Language: English | DSM C-DIS | a) COI declaration 'None disclosed' b) Funding acknowledged (academic/health research institutions) |
| Thompson et al[39] | Country: USA Setting: patients with Parkinson's disease Age (years): 72.5 (SD=9.6) Female: 42% | n=214 Depressed: 14% | Administration:self administered Language: English | DSM-IV SCID | a) No COI declaration b) Funding acknowledged (academic/health research institutions) |
| Turner et al[40] | Country: Australia Setting: stroke patients Age (years): male=66.7 (SD=13.1) Female: 47.2% | n=72 Depressed: 18% | Administration:self administered Language: English | DSM-IV SCID | a) COI declaration: disclosures 'none'. b) Funding acknowledged (academic/health research institutions) |
| van Steenbergen-Weijenburg et al[41] | Country: The Netherlands Setting: patients with diabetes Age (years): male=61.8 (SD=13.6) Female: 48.7% | n=197 Depressed: 18.8% | Administration: self administered Language: Dutch | DSM-IV SCID | a) COI declaration: 'The authors declare that they have no competing interests'. b) Funding acknowledged (academic/health research institutions)—'this had no influence on the content of this article'. |
| Zuithoff et al[47] | Country: The Netherlands Setting: primary care Age (years): male=51 (SD=16.7) Female: 63% | n=1338 Depressed: 13% | Administration: self-report Language: Dutch | DSM-IV CIDI | a) COI declaration 'The authors declare that they have no competing interests'. b) Funding acknowledged (academic/health research institutions) |

CIDI, Composite International Diagnostic Interview;CIS-R, Clinical Interview Schedule;COI, conflict of interest; DSM, Diagnostic and Statistical Manual of Mental Disorders; MINI, Mini-International Neuropsychiatric Interview; N/A, not available; SCAN, Schedules for Clinical Assessments in Neuropsychiatry; SCID, Structured Clinical Interview for DSM Disorders.

**Table 2** Descriptive characteristics of the summed items scoring method studies cut-off point 10[14]

| Study | Sample characteristics | Sample size and % MDD | PHQ-9 characteristics | Diagnostic standard | a) COI declaration b) Funding c) Relationship with original developers |
|---|---|---|---|---|---|
| 13. Gräfe et al[23] | Country: Germany Setting: psychosomatic walk-in clinics and family practices Mean age: 41.9 (SD=13.8) Female: 67.8% | n=528 Depressed: 29.2% psychosomatic patients; 6.16% medical patients | Administration: self-report Language: German Cut-offs: 10–14 | DSM-IV SCID | a) No COI declaration b) Acknowledged funding from Pfizer c) Not acknowledged |
| 16. Kroenke et al[12] | Country: USA Setting: primary care Mean age: 46 (SD=17) Female: 66% | n=580 7.1% MDD | Administration: self-report Language: English Cut-offs: 9–15 | DSM-IV SCID | a) No COI declaration b) Acknowledged funding from Pfizer c) N/A |
| 22. Navinés et al[28] | Country: Spain Setting: general hospital (patients with chronic HCV) Mean age: 43.4 (SD=10.2) Female: 28.6% | n=500 6.4% MDD | Administration: self-report Language: Spanish Cut-offs: 10 | DSM-IV SCID | a) All authors declared that they had no COI b) Role of funding source declared c) Not acknowledged |
| 29. Thekkumpurath et al[26] | Country: UK Setting: hospital (cancer patients) Mean age: 61 Female: 63% | n=782 6.3% MDD (of the whole sample) | Administration: not stated Language: English Cut-offs: 5–10 | DSM-IV SCID | a) COI declaration: 'Supported by Cancer Research UK' b) As in a) c) Not acknowledged |
| 33. Williams et al[49] | Country: USA Setting: secondary care (poststroke) Mean age: unclear Female: unclear | n=316 33.5% MDD | Administration: unclear Language: English Cut-offs: 10 | DSM-IV SCID | a) No COI declaration b) Funding acknowledged (academic institutions) c) Not acknowledged |
| 1. Adewuya et al[55] | Country: Nigeria Setting: community (students) Mean age: 24.8 (15–40) Female: 41.2% | n=512 2.5% MDD | Administration: Self-report Language: English Cut-offs: 8–12 | DSM-IV MINI | a) No COI declaration b) No funding declaration |
| 2. Arroll et al[42] | Country: New Zealand Setting: primary care Mean age: 49 (17–99) Female: 61% | n=2642 6.2% MDD | Administration: not stated Language: English Cut-offs: 8, 10, 12, 15 | DSM-IV SCID | a) No COI declaration b) Funding acknowledged (academic/health research institutions) |
| 3. Azah et al[62] | Country: Malaysia Setting: primary care Mean age: 38.7 (18–79) Female: 61.7% | n=180 16.6% MDD | Administration: self-report Language: Malay Cut-offs: 5–12 | DSM-IV CIDI | b) No COI declaration c) Funding acknowledged (academic/health research institutions) |
| 4. Chagas et al[50] | Country: Brazil Setting: secondary care Mean age: not stated Female: 52.7% | n=84 25.5% MDD | Administration: self-report Language: Brazilian Cut-offs: 7–10 | DSM-IV SCID | a) COI declaration 'None declared' b) Funding acknowledged (academic/health research institutions) |

Continued

**Table 2** Continued

| Study | Sample characteristics | Sample size and % MDD | PHQ-9 characteristics | Diagnostic standard | a) COI declaration<br>b) Funding<br>c) Relationship with original developers |
|---|---|---|---|---|---|
| 6. de Lima Osorio et al[60] | Country: Brazil<br>Setting: primary care<br>Mean age: unclear<br>Female: 100% | n=177<br>34% MDD | Administration: research assistants<br>Language: Brazilian Portuguese<br>Cut-offs: 10–15 | DSM-IV<br>SCID | a) No COI declaration<br>b) Funding acknowledged (academic institutions) |
| 7. Elderon et al[51] | Country: USA<br>Setting: secondary care<br>Mean age: unclear<br>Female: 18% | n=1022<br>18.3% MDD | Administration: self-report<br>Language: English<br>Cut-offs: 10 | C-DIS | a) COI declaration—'No disclosures'<br>b) Funding acknowledged (academic institutions and industry—AHA Pharmaceuticals Roundtable)—'The funding organisations had no role in the design or conduct of the study, collection, management, analysis or interpretation of data; or preparation, review or approval of the manuscript'. |
| 8. Fann et al[30] | Country: USA<br>Setting: trauma hospital (inpatients with traumatic brain injury)<br>Mean age: 42 (SD=17.9)<br>Female: 29.1% | n=135<br>16.3% MDD | Administration: telephone-administered<br>Language: English<br>Cut-offs: 10 | DSM-IV<br>SCID | a) No COI declaration<br>b) Funding acknowledged (academic institutions) |
| 9. Fine et al[56] | Country: USA<br>Setting: primary care (Ohio Army National Guard)<br>Mean age: 31 (17–60)<br>Female: 12% | n=498<br>21.5% MDD | Administration: telephone-administered<br>Language: English<br>Cut-offs: 10, 15 | DSM-IV<br>SCID-I | a) COI—last author disclosed financial and consulting interests (Pfizer not one of them). All other authors declared that they have no COI.<br>b) Funding acknowledged—DoD Medical Research. 'The sponsor had no role in study design, data collection, analysis, interpretation of results, report writing or manuscript submission'. |
| 10. Gelaye et al[31] | Country: Ethiopia<br>Setting: general hospital<br>Mean age: 34.9 (SD=11.6)<br>Female: 63.1% | n=363<br>12.6% MDD | Administration: researcher-administered<br>Language: Amharic<br>Cut-offs: 9–11 | DSM-IV<br>SCAN | a) No COI declaration<br>b) Funding acknowledged (academic/health research institutions) |
| 11. Gilbody et al[57] | Country: UK<br>Setting: primary care<br>Mean age: 42.5 (SD 13.6)<br>Female: 77% | n=96<br>37.5 MDD | Administration: not stated<br>Language: English<br>Cut-offs: 9–13 | DSM-IV<br>SCID | a) COI declaration—last author involved in the development of one of the instruments (CORE-OM), 'but does not gain financially from its use.<br>b) Funding acknowledged (academic/health research institutions) |
| 12. Gjerdingen et al[48] | Country: USA<br>Setting: community<br>Mean age: 29.3<br>Female: 100% | n=438<br>4.6% MDD | Administration: telephone or self-report<br>Language: English<br>Cut-offs: 10 | DSM-IV<br>SCID | c) No COI declaration<br>d) Funding acknowledged (academic/health research institutions) |
| 14. Hyphantis et al[32] | Country: Greece<br>Setting: hospital— rheumatology patients<br>Mean age: 54.2 (SD=13.5)<br>Female: 74% | n=213<br>32.4% MDD | Administration: researcher administered<br>Language: Greek<br>Cut-offs: 4–16 | DSM-IV<br>MINI | a) No COI declaration<br>b) No funding acknowledgement |

Continued

**Table 2** Continued

| Study | Sample characteristics | Sample size and % MDD | PHQ-9 characteristics | Diagnostic standard | a) COI declaration<br>b) Funding<br>c) Relationship with original developers |
|---|---|---|---|---|---|
| 15. Khamseh et al[34] | Country: Iran<br>Setting: outpatient diabetic clinic<br>Mean age: 56.1 (SD=9.6)<br>Female: 51.8% | n=185<br>43.2% MDD | Administration: self-report<br>Language: Persian<br>Cut-offs: 10, 13 | DSM-IV<br>SCID | a) COI declaration: the authors declared no competing interests.<br>d) Funding acknowledged (academic/health research institutions) |
| 19. Liu et al[63] | Country: Taiwan<br>Setting: primary care<br>Mean age: not specified<br>Female: 60.9% | n=1532<br>3.3% MDD | Administration: self-report<br>Language: Chinese version<br>Cut-offs: 9–11 | SCAN | a) No COI declaration<br>b) Funding acknowledged (academic/healthresearch institutions) |
| 20. Lotrakul et al[46] | Country: Thailand<br>Setting: primary care<br>Mean age: 45.0 (SD=14.30)<br>Female: 73.7% | n=279<br>6.8% MDD | Administration: self report<br>Language: Thai<br>Cut-offs: 7–15 | DSM-IV<br>MINI | a) No COI declaration<br>d) Funding acknowledged (academic/healthresearch institutions) |
| 23. Patel et al[61] | Country: India<br>Setting: primary care<br>Mean age: 37.5 (18–83)<br>Female: 56.4% | n=299<br>4.3% MDD | Administration: face-to-face interview<br>Language: not specified<br>Cut-offs: 7–15 | CIS-R | a) COI declaration—No declaration of Interest<br>b) Funding acknowledged (academic/healthresearch institutions) |
| 24. Phelan et al[58] | Country: USA<br>Setting: primary care (elderly)<br>Mean age: 78 (SD=7)<br>Female: 62% | n=71<br>12% MDD | Administration: research assistant<br>Language: English<br>Cut-offs: 8–12 | DSM-IV<br>SCID | a) COI declaration—no competing interests<br>b) Funding acknowledged (academic/healthresearch institutions). 'The funder had no role in the study design, methods, data collection, analysis or interpretation of data, nor any role in the preparation of the manuscript or decision to submit the manuscript for publication'. |
| 25. Rooney et al[52] | Country: UK<br>Setting: secondary care (glioma)<br>Mean age: 54.2 (SD=12.3)<br>Female: 42.6% | n=129<br>13.5% MDD | Administration: self-report<br>Language: English<br>Cut-offs: 8–11 | DSM-IV<br>SCID | a) COI declaration 'The authors declare that they have no COI'.<br>b) Funding acknowledged (academic/health research institutions) |
| 26. Sherina et al | Country: Malaysia<br>Setting: primary care<br>Mean age: 30.9 (18–81)<br>Female: 100% | n=146<br>21.2% MDD | Administration: self-report<br>Language: Malay<br>Cut-offs: 10 | CIDI | a) COI declaration 'The authors declare that they have no competing interests'.<br>b) Funding acknowledged (academic/health research institutions) |
| 27. Sidebottom et al[59] | Country: USA<br>Setting: community (prenatal)<br>Mean age: 23 (SD=5.5)<br>Female: 100% | n=745<br>3.6% MDD | Administration: interview<br>Language: English<br>Cut-offs: 10 | DSM-IV<br>SCID | b) COI declaration 'The authors declare that they have no financial COI'.<br>b) Funding acknowledged (academic/health research institutions) |
| 28. Stafford et al[37] | Country: Australia<br>Setting: secondary care (cardiac procedures)<br>Mean age: 64.14 (38–91)<br>Female: 19.2% | n=193<br>18.1% MDD | Administration: self-report<br>Language: English<br>Cut-offs: 10 | DSM-IV MINI | a) No COI declaration<br>b) Funding acknowledged (academic/health research institutions) |

Continued

**Table 2** Continued

| Study | Sample characteristics | Sample size and % MDD | PHQ-9 characteristics | Diagnostic standard | a) COI declaration<br>b) Funding<br>c) Relationship with original developers |
|---|---|---|---|---|---|
| 30. Thombs et al[38] | Country: USA<br>Setting: hospital (outpatients with coronary heart disease)<br>Mean age: 67 (SD=11)<br>Female: 18% | n=1024<br>22% MDD | Administration: not stated<br>Language: English<br>Cut-offs: 7–10 | DSM<br>C-DIS | a) COI declaration 'None disclosed'<br>b) Funding acknowledged (academic/health research institutions) |
| 32. Watnick et al[53] | Country: USA<br>Setting: secondary care (dialysis)<br>Mean age: 63 (SD=15)<br>Female: 32.3% | n=62<br>19% MDD | Administration: self-report<br>Language: English<br>Cut-offs: 10 | DSM-IV<br>SCID | a) No COI declaration<br>b) Funding acknowledged (academic/health research institutions) |
| 34. Wittkampf et al[64] | Country: The Netherlands<br>Setting: primary care<br>Mean age: 49.8<br>Female: 66.7% | n=664<br>12.3% MDD | Administration: self-report<br>Language: not specified<br>Cut-offs: 10 and 15 | DSM-IV<br>SCIDI | a) No COI declaration<br>b) Funding acknowledged (academic/health research institutions) |
| 35. Zhang et al[54] | Country: Hong Kong<br>Setting: secondary care (diabetic outpatients)<br>Mean age: 55.1 (SD=9.5)<br>Female: 40.8% | n=99<br>23.2% MDD | Administration: self-report<br>Language: Chinese version<br>Cut-offs: 15 | DSM-IV MINI | a) COI declaration—last author acknowledged financial COI. The other authors declare that they have no competing interests.<br>b) Funding acknowledged (academic/health research institutions) |
| 36. Zuithoff et al[47] | Country: The Netherlands<br>Setting: primary care<br>Age (years): male=51 (SD=16.7)<br>Female: 63% | n=1338<br>Depressed: 13% | Administration: self-report<br>Language: Dutch | DSM-IV<br>CIDI | a) COI declaration 'The authors declare that they have no competing interests'.<br>b) Funding acknowledged (academic/health research institutions) |

COI, conflict of interest; DSM, Diagnostic and Statistical Manual of Mental Disorders; MDD, major depressive disorder; N/A, not available; SCID, Structured Clinical Interview for DSM Disorders.

**Table 3** Quality assessment of included studies in the algorithm meta-analysis[13]

| Study | Patient selection: Consecutive or random sample | Patient selection: Avoid case-control/avoid artificially inflated base rate | Patient selection: Avoided inappropriate exclusions | Patient selection: Overall risk of bias | Index test: PHQ-9 interpreted blind to reference test | Index test: If translated, appropriate translation | Index test: If translated, psychometric properties reported | Index test: Overall risk of bias |
|---|---|---|---|---|---|---|---|---|
| **Allegiant studies** | | | | | | | | |
| Diez-Quevedo et al[22] | ✗ | ✓ | ✗ | High | ? | ✓ | ✓ | Unclear |
| Gräfe et al[23] | ✓ | ✓ | ✓ | Low | ? | ✓ | ✓ | Unclear |
| Lowe et al[24] | ✗ | ✓ | ✓ | High | ✓ | ✓ | ✓ | Low |
| Muramatsu et al[27] | ? | ✓ | ? | Unclear | ✓ | ✓ | ? | Unclear |
| Navines et al[28] | ✓ | ✓ | ✓ | Low | ✓ | ✓ | ? | Unclear |
| Spitzer et al[25] | ✗ | ✓ | ✓ | High | ✓ | N/A | N/A | Low |
| Thekkumpurath et al[26] | ✗ | ✗ | ✓ | High | ✓ | N/A | N/A | Low |
| **Non-allegiant studies** | | | | | | | | |
| Arroll et al[42] | ✓ | ✓ | ✓ | Low | ✓ | N/A | N/A | Low |
| Ayalon et al[43] | ? | ✓ | ✓ | Unclear | ? | ✓ | ? | Unclear |
| Eack et al[29] | ? | ✓ | ? | Unclear | ? | N/A | N/A | Unclear |
| Fann et al[30] | ✓ | ✗ | ✗ | High | ✓ | N/A | N/A | Low |
| Gelaye et al[31] | ? | ✗ | ? | High | ✓ | ✓ | ? | Unclear |
| Gjerdingen et al[48] | ✓ | ✓ | ✓ | Low | ? | N/A | N/A | Unclear |
| Henkel et al[44] | ✓ | ✓ | ✓ | Low | ? | N/A | N/A | Unclear |
| Hyphantis et al[32] | ✓ | ✓ | ✗ | High | ✓ | ? | ? | Unclear |
| Inagaki et al[33] | ✓ | ✗ | ✓ | High | ✓ | ? | ? | Unclear |
| Khamseh et al[34] | ✓ | ✓ | ? | Unclear | ✓ | ✓ | ? | Unclear |
| Lamers et al[45] | ✓ | ✗ | ✗ | High | ✓ | ? | ? | Unclear |
| Lotrakul et al[46] | ✗ | ✓ | ? | High | ? | ✓ | ? | Unclear |
| Persoons et al[35] | ✓ | ✓ | ✓ | Low | ✓ | ✓ | N/A | Low |
| Picardi et al[36] | ✓ | ✓ | ✓ | Low | ✓ | ? | ? | Unclear |
| Stafford et al[37] | ✓ | ✓ | ✓ | Low | ✓ | N/A | N/A | Low |
| Thombs et al[38] | ✗ | ✓ | ? | Unclear | ? | N/A | N/A | Unclear |
| Thomspon et al[39] | ? | ✓ | ✓ | Unclear | ? | N/A | N/A | Unclear |
| Turner et al[40] | ✓ | ✓ | ✓ | Low | ✓ | N/A | N/A | Low |

Continued

**Table 3** Continued

| Study | Patient selection: Consecutive or random sample | Patient selection: Avoid case-control/avoid artificially inflated base rate | Patient selection: Avoided inappropriate exclusions | Patient selection: Overall risk of bias | Reference test: Reference test correctly classifies target condition | Reference test: Reference test interpreted blind to PHQ-9 | Reference test: If translated, appropriate translation | Reference test: If translated, psychometric properties reported | Reference test: Overall risk of bias | Index test: PHQ-9 interpreted blind to reference test | Index test: If translated, appropriate translation | Index test: If translated, psychometric properties reported | Index test: Overall risk of bias | Flow/timing: Interval of 2 weeks or less | Flow/timing: All participants receive same reference test | Flow/timing: All participants included in analysis? | Flow/timing: Overall risk of bias |
|---|---|---|---|---|---|---|---|---|---|---|---|---|---|---|---|---|---|
| van Steenbergen-Wijenburg et al[41] | ? | ✓ | ✓ | Unclear | ✓ | ✓ | ✓ | ? | Unclear | ? | ? | ? | Unclear | | | | |
| Zuithoff et al[47] | ✓ | ✓ | ✓ | Low | ✓ | ✓ | ✓ | ? | Low | ✓ | ✓ | ? | Unclear | | | | |
| **Allegiant studies** | | | | | | | | | | | | | | | | | |
| Diez-Quevedo et al[22] | | | | | ✓ | ✓ | ✓ | ? | Unclear | ✓ | ✓ | ✓ | Low | ✓ | ✓ | ✓ | |
| Gräfe et al[23] | | | | | ✓ | ? | N/A | N/A | Unclear | ✓ | ✓ | ✓ | Low | ✓ | ✓ | ✓ | |
| Lowe et al[24] | | | | | ✓ | ✓ | N/A | N/A | Low | ✓ | ✓ | ✓ | Low | ✓ | ✓ | ✓ | |
| Muramatsu et al[27] | | | | | ✓ | ✓ | ✓ | ✓ | Low | ✓ | ✓ | ? | Unclear | ✓ | ✓ | ? | |
| Navines et al[28] | | | | | ✓ | ✓ | ? | ? | Unclear | ✓ | ✓ | ✓ | Low | ✓ | ✓ | ✓ | |
| Spitzer et al[25] | | | | | ✓ | ✓ | N/A | N/A | Low | ✓ | ✓ | ✗ | High | ✓ | ✓ | ✗ | |
| Thekkumpurath et al[26] | | | | | ✓ | ✓ | N/A | N/A | Low | ? | ✓ | ✗ | High | ? | ✓ | ✗ | |
| **Non-allegiant studies** | | | | | | | | | | | | | | | | | |
| Arroll et al[42] | | | | | ✓ | ✓ | N/A | N/A | Low | ✓ | ✓ | ✓ | Low | ✓ | ✓ | ✓ | |
| Ayalon et al[43] | | | | | ✓ | ? | ✓ | ? | Unclear | ? | ✓ | ✓ | Unclear | ? | ✓ | ✓ | |
| Eack et al[29] | | | | | ✓ | ? | N/A | N/A | Unclear | ? | ✓ | ? | Unclear | ? | ✓ | ? | |
| Fann et al[30] | | | | | ✓ | ? | N/A | N/A | Unclear | ✓ | ✓ | ✗ | High | ✓ | ✓ | ✗ | |
| Gelaye et al[31] | | | | | ✓ | ✓ | ✓ | ✓ | Low | ✓ | ✓ | ✗ | High | ✓ | ✓ | ✗ | |
| Gjerdingen et al[48] | | | | | ✓ | ? | N/A | N/A | Unclear | ✓ | ✓ | ✗ | High | ✓ | ✓ | ✗ | |
| Henkel et al[44] | | | | | ✓ | ? | N/A | N/A | Unclear | ✓ | ✓ | ✗ | High | ✓ | ✓ | ✗ | |
| Hyphantis et al[32] | | | | | ✓ | ✓ | ? | ? | Unclear | ✓ | ✓ | ✗ | High | ✓ | ✓ | ✗ | |
| Inagaki et al[33] | | | | | ✓ | ✓ | ✓ | ? | Unclear | ✓ | ✓ | ✗ | High | ✓ | ✓ | ✗ | |
| Khamseh et al[34] | | | | | ✓ | ✓ | ✓ | ? | Unclear | ✓ | ✓ | ? | Unclear | ✓ | ✓ | ? | |
| Lamers et al[45] | | | | | ✓ | ✓ | ? | ? | Unclear | ? | ✓ | ✗ | High | ? | ✓ | ✗ | |

Continued

**Table 3** Continued

| Study | Reference test: Reference test correctly classifies target condition | Reference test: Reference test interpreted blind to PHQ-9 | Reference test: If translated, appropriate translation | Reference test: If translated, psychometric properties reported | Reference test: Overall risk of bias | Flow/timing: Interval of 2 weeks or less | Flow/timing: All participants receive same reference test | Flow/timing: All participants included in analysis? | Flow/timing: Overall risk of bias |
|---|---|---|---|---|---|---|---|---|---|
| Lotrakul et al[46] | ✓ | ✓ | ✓ | ✓ | Low | ? | ✓ | ✗ | High |
| Persoons et al[35] | ✓ | ✓ | ? | ? | Unclear | ✓ | ✓ | ✓ | Low |
| Picardi et al[36] | ✓ | ✓ | ✓ | ? | Unclear | ✓ | ✓ | ✗ | High |
| Stafford et al[37] | ✓ | ✓ | N/A | N/A | Low | ✓ | ✓ | ✗ | High |
| Thombs et al[38] | ? | ✓ | N/A | N/A | Unclear | ✓ | ✓ | ✓ | Low |
| Thompson et al[39] | ✓ | ? | N/A | N/A | Unclear | ✓ | ✓ | ✗ | High |
| Turner et al[40] | ✓ | ? | N/A | N/A | Unclear | ? | ✓ | ✗ | High |
| van Steenbergen-Wijenburg et al[41] | ✓ | ✗ | ? | ? | High | ✓ | ✓ | ✗ | High |
| Zuithoff et al[47] | ✓ | ✓ | ? | ? | Unclear | ? | ✓ | ✓ | Unclear |

✓, criterion met; ✗, criterion not met; ?, insufficient information to code whether criterion met; N/A, not applicable; PHQ-9, Patient Health Questionnaire-9.

**Table 4** Quality assessment of included studies in the summed item scoring method cut-off point 10 meta-analysis[14]

| Study | Patient selection: Consecutive or random sample | Patient selection: Avoid case-control/avoid artificially inflated base rate | Patient selection: Avoided inappropriate exclusions | Patient selection: Overall risk of bias | Index test: PHQ-9 interpreted blind to reference test | Index test: Was a threshold prespecified? | Index test: If translated, appropriate translation | Index test: If translated, psychometric properties reported | Index test: Overall risk of bias |
|---|---|---|---|---|---|---|---|---|---|
| **Allegiant studies** | | | | | | | | | |
| 13. Gräfe et al[23] | ✓ | ✓ | ✓ | Low | ? | ✓ | ✓ | ✓ | Unclear |
| 16. Kroenke et al[12] | ✓ | ✓ | ✓ | Low | ✓ | ✓ | N/A | N/A | Low |
| 22. Navinés et al[28] | ✓ | ✓ | ✓ | Low | ✓ | ✓ | ✓ | ? | Unclear |
| 29. Thekkumpurath et al[26] | × | × | ✓ | High | ✓ | ✓ | N/A | N/A | Low |
| 33. Williams et al[49] | ✓ | ✓ | ✓ | Low | ? | ✓ | N/A | N/A | Unclear |
| **Non-allegiant studies** | | | | | | | | | |
| 1. Adewuya et al[55] | ✓ | ✓ | × | Unclear | ✓ | ✓ | N/A | N/A | Low |
| 2. Arroll et al[42] | ✓ | ✓ | ✓ | Low | ✓ | ✓ | N/A | N/A | Low |
| 3. Azah et al[62] | ✓ | × | ? | High | ✓ | ✓ | ✓ | ✓ | Low |
| 4. Chagas et al[50] | ✓ | ✓ | ✓ | Low | ✓ | ✓ | ✓ | ✓ | Low |
| 6. de Lima Osorio et al[60] | ✓ | × | ✓ | High | ? | × | N/A | N/A | High |
| 7. Elderon et al[51] | ✓ | ✓ | ✓ | Low | ✓ | ✓ | N/A | N/A | Low |
| 8. Fann et al[30] | ✓ | × | × | High | ✓ | ✓ | N/A | N/A | Low |
| 9. Fine et al[56] | ✓ | ✓ | ✓ | Low | ? | ✓ | N/A | N/A | Unclear |
| 10. Gelaye et al[31] | ? | × | ? | High | ✓ | × | ✓ | ? | High |
| 11. Gilbody et al[57] | ? | ✓ | ? | Unclear | ✓ | ✓ | N/A | N/A | Low |
| 12. Gjerdingen et al[48] | ✓ | ✓ | ✓ | Low | ? | ✓ | N/A | N/A | Unclear |
| 14. Hyphantis et al[32] | ✓ | × | ✓ | High | ✓ | ✓ | ? | ? | Unclear |
| 15. Khamseh et al[34] | ✓ | ✓ | ? | Unclear | ✓ | ✓ | ✓ | ? | Unclear |
| 19. Liu et al[63] | ✓ | ✓ | ? | Unclear | ✓ | × | ✓ | ? | High |
| 20. Lotrakul et al[46] | × | ✓ | ? | Unclear | ✓ | ✓ | ✓ | ? | Unclear |
| 23. Patel et al[61] | ✓ | ✓ | ✓ | Low | ✓ | ✓ | ? | ? | Unclear |
| 24. Phelan et al[58] | × | ✓ | ✓ | High | ✓ | × | N/A | N/A | High |
| 25. Rooney et al[52] | ✓ | ✓ | ✓ | Low | ? | × | N/A | N/A | High |
| 26. Sherina et al | ✓ | ✓ | × | High | ✓ | ✓ | ✓ | ✓ | Low |
| 27. Sidebottom et al[59] | ✓ | ✓ | ✓ | Low | ✓ | ✓ | N/A | N/A | Low |
| 28. Stafford et al[37] | ✓ | ✓ | ✓ | Low | ✓ | ✓ | N/A | N/A | Low |

**Table 4** Continued

| Study | Patient selection: Consecutive or random sample | Patient selection: Avoid case-control/avoid artificially inflated base rate | Patient selection: Avoided inappropriate exclusions | Patient selection: Overall risk of bias | Index test: PHQ-9 interpreted blind to reference test | Index test: Was a threshold prespecified? | Index test: If translated, appropriate translation | Index test: If translated, psychometric properties reported | Index test: Overall risk of bias |
|---|---|---|---|---|---|---|---|---|---|
| 30. Thombs et al[38] | × | ✓ | ? | High | ✓ | ? | N/A | N/A | Unclear |
| 32. Watnick et al[53] | ? | × | ✓ | High | ✓ | ✓ | N/A | N/A | Low |
| 34. Wittkampf et al[64] | ✓ | ✓ | ✓ | Low | ✓ | ? | N/A | N/A | Unclear |
| 35. Zhang et al[54] | ✓ | ✓ | ? | Unclear | ? | ✓ | ? | ? | Unclear |
| 36. Zuithoff et al[47] | ✓ | ✓ | ✓ | Low | ✓ | ✓ | ✓ | ? | Unclear |

| Study | Reference test: Reference test correctly classifies target condition | Reference test: Reference test interpreted blind to PHQ-9 | Reference test: If translated, appropriate translation | Reference test: If translated, psychometric properties reported | Reference test: Overall risk of bias | Flow / timing: Interval of 2 weeks or less | Flow / timing: All participants receive same reference test | Flow / timing: All participants included in analysis? | Flow / timing: Overall risk of bias |
|---|---|---|---|---|---|---|---|---|---|
| **Allegiant studies** | | | | | | | | | |
| 13. Gräfe et al[23] | ✓ | ? | N/A | N/A | Unclear | ✓ | ✓ | ✓ | Low |
| 16. Kroenke et al[12] | ✓ | ✓ | N/A | N/A | Low | ✓ | ✓ | ✓ | Low |
| 22. Navinés et al[28] | ✓ | ✓ | ? | ? | Unclear | ✓ | ✓ | ✓ | Low |
| 29. Thekkumpurath et al[26] | ✓ | ✓ | N/A | N/A | Low | ? | ✓ | ✓ | Unclear |
| 33. Williams et al[49] | ✓ | ? | N/A | N/A | Unclear | ? | ✓ | ✓ | Unclear |
| **Non-allegiant studies** | | | | | | | | | |
| 1. Adewuya et al[55] | ✓ | ✓ | N/A | N/A | Low | ✓ | ✓ | ✓ | Low |
| 2. Arroll et al[42] | ✓ | ✓ | N/A | N/A | Low | ? | ✓ | ✓ | Unclear |
| 3. Azah et al[62] | ✓ | ✓ | ✓ | ✓ | Low | ✓ | ✓ | × | High |
| 4. Chagas et al[50] | ✓ | ✓ | ? | ? | Unclear | ✓ | ✓ | × | High |
| 6. de Lima Osorio et al[60] | ✓ | ? | N/A | N/A | Unclear | ? | ✓ | ✓ | Unclear |
| 7. Elderon et al[51] | ✓ | ✓ | N/A | N/A | Low | ✓ | ✓ | ✓ | Low |
| 8. Fann et al[30] | ✓ | ? | N/A | N/A | Unclear | ✓ | ✓ | × | High |
| 9. Fine et al[56] | ✓ | ? | N/A | N/A | Unclear | ? | ✓ | ✓ | Unclear |
| 10. Gelaye et al[31] | ✓ | ✓ | ✓ | ✓ | Low | ✓ | ✓ | × | High |
| 11. Gilbody et al[57] | ✓ | ✓ | N/A | N/A | Low | ? | ✓ | ✓ | Unclear |
| 12. Gjerdingen et al[48] | ✓ | ? | N/A | N/A | Unclear | ✓ | ✓ | × | High |

Continued

**Table 4** Continued

| Study | Reference test: Reference test correctly classifies target condition | Reference test: Reference test interpreted blind to PHQ-9 | Reference test: If translated, appropriate translation | Reference test: If translated, psychometric properties reported | Reference test: Overall risk of bias | Flow / timing: Interval of 2 weeks or less | Flow / timing: All participants receive same reference test | Flow / timing: All participants included in analysis? | Flow / timing: Overall risk of bias |
|---|---|---|---|---|---|---|---|---|---|
| 14. Hyphantis et al[32] | ✓ | ✓ | ? | ? | Unclear | ✓ | ✓ | ✗ | High |
| 15. Khamseh et al[34] | ✓ | ✓ | ✓ | ? | Unclear | ✓ | ✓ | ? | Unclear |
| 19. Liu et al[63] | ✓ | ✓ | ✓ | ✓ | Low | ✓ | ✓ | ? | Unclear |
| 20. Lotrakul et al[46] | ✓ | ✓ | ✓ | ✓ | Low | ? | ✓ | ✗ | High |
| 23. Patel et al[61] | ✓ | ✓ | ✓ | ? | Unclear | ? | ✓ | ✗ | High |
| 24. Phelan et al[58] | ✓ | ✓ | N/A | N/A | Low | ✓ | ✓ | ✓ | Low |
| 25. Rooney et al[62] | ✓ | ? | N/A | N/A | Unclear | ? | ✓ | ✗ | High |
| 26. Sherina et al | ✓ | ✓ | ✓ | ✓ | Low | ✓ | ✓ | ✓ | Low |
| 27. Sidebottom et al[59] | ✓ | ✓ | N/A | N/A | Low | ✓ | ✓ | ✗ | High |
| 28. Stafford et al[37] | ✓ | ✓ | N/A | N/A | Low | ✓ | ✓ | ✗ | High |
| 30. Thombs et al[38] | ? | ✓ | N/A | N/A | Unclear | ✓ | ✓ | ✓ | Low |
| 32. Watnick et al[53] | ✓ | ✓ | N/A | N/A | Low | ✓ | ✓ | ✓ | Low |
| 34. Wittkampf et al[64] | ✓ | ✓ | N/A | N/A | Low | ? | ✓ | ✗ | High |
| 35. Zhang et al[54] | ✓ | ? | ✓ | ✓ | Unclear | ✗ | ✓ | ✗ | High |
| 36. Zuithoff et al[47] | ✓ | ✓ | ? | ? | Unclear | ? | ✓ | ✓ | Unclear |

N/A, not applicable; PHQ-9, Patient Health Questionnaire-9. ✓, criterion met; ✗, criterion not met; ?, insufficient information to code whether criterion met.

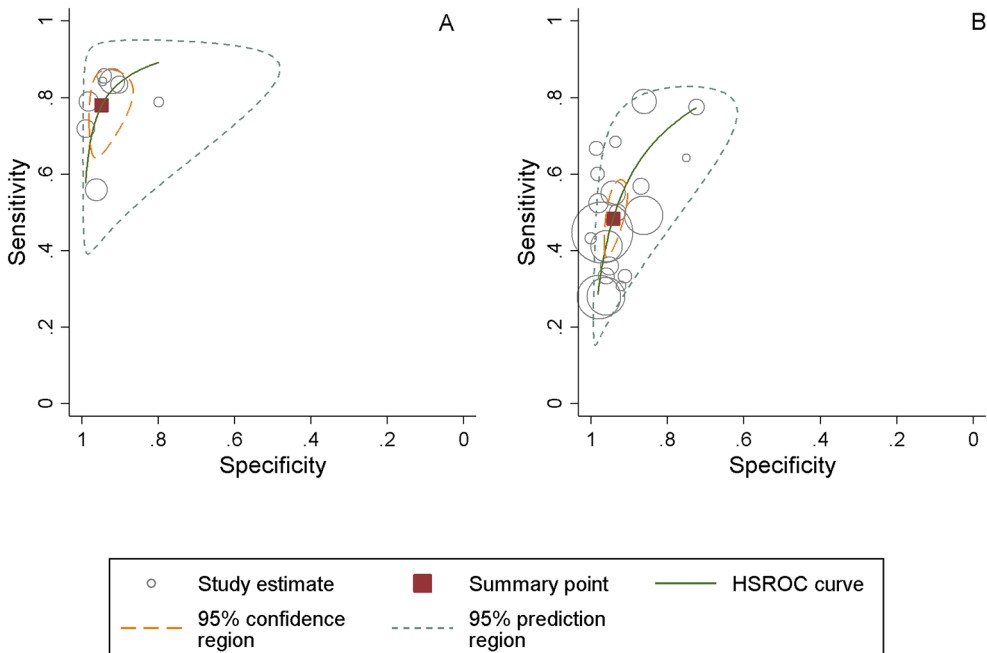

**Figure 1** Patient Health Questionnaire-9 algorithm scoring method summary receiver operating characteristic plot for the diagnosis of major depressive disorder in allegiant studies (panel A) and non-allegiant studies (panel B). Pooled sensitivity and specificity estimates using a bivariate meta-analysis. HSROC, hierarchical receiver operating characteristic.

case-control design, avoid inappropriate exclusions). In the index test domain, the proportion of studies reporting that the PHQ-9 was conducted blind to the reference test was comparable between the two groups. There were differences in this domain for those studies using a translated version of the test. All non-English allegiant studies (5/5) used an appropriately translated version of the PHQ-9, whereas just over a half of the non-allegiant studies reported this (55%, 6/11). However, the majority of both sets of studies did not report details of psychometric properties of the translated version. For the reference test domain, nearly all studies in both groups were rated as using a reference test that would correctly classify the condition. While most allegiant studies reported that the reference test was interpreted blind to the PHQ-9 score (86%, 6/7), this was reported in only 60% (12/20) of the non-allegiant studies.

The two sets of studies that used translated versions of the reference test were broadly comparable. There was a slight indication that the allegiant studies were more likely to use an appropriately translated version of the reference test and report data on the psychometric properties of the translated version, although the numbers for the translated comparison are very low. There were, however, some more notable differences on the flow and timing domain. Most allegiant studies ensured that the time between the index and reference test was under 2 weeks (86%, 6/7) in comparison to 70% (14/20) of the non-allegiant studies. More allegiant studies met the criterion for 'all participants included in the analysis' (57%, 4/7) than non-allegiant studies (25%).

### Summed items scoring method (cut-off point 10 or above)
#### Descriptive characteristics

Table 2 presents the sample characteristics of the 31 PHQ-9 validation studies that reported the psychometric properties of the PHQ-9 at cut-off point 10 or above. Five of these studies were coauthored by the original developers of the instrument or acknowledged collaboration[12 23 26 49] or were coauthored by the first author of a previous study that had also been coauthored by one of the developers.[28] Twenty-six studies were conducted by independent researchers.

Three (60%, 3/5) allegiant studies[26 28 49] and 11 non-allegiant studies (42%, 11/26)[30–32 34 37 38 50–54] were conducted in hospital settings.

Three (60%, 3/5) allegiant studies[12 26 49] and 13 non-allegiant studies (13/26)[30 37 38 42 48 51–53 55–59] were conducted in English.

The mean prevalence of MDD in the allegiant group was 13.2% (range 6.1%–33.5%) and in the non-allegiant group was 16.1% (range 2.5%–43.2%). The mean age of patients in the allegiant group studies was 48.1 (range 41.9–61.0) and in the 26 non-allegiant studies that reported these data was 49.1 (range 23.0–78.0). The percentage of females in the allegiant studies that reported these data[12 23 26 28] was 56.3% (range 28.6%–67.8%) and in the non-allegiant group was 64.9% (range 12%–100%).

Three allegiant studies used the self-reported mode of administration and two of them did not specify how the PHQ-9 was administered. In nine non-allegiant studies (34%, 9/26), the PHQ-9 was administered by the researcher.[30–32 48 56 58–61] All allegiant studies used SCID as

a gold standard; the non-allegiant studies used a wider range of gold standards including SCAN, CIDI, MINI, CIS-R, C-DIS, although the SCID was used in half of the studies (50%, 13/26 studies).

Three allegiant studies (60%) did not include a conflict of interest statement.[12 23 49] Two of these studies[12 23] acknowledged funding from Pfizer. None of the allegiant studies acknowledged collaboration or authorship of one of the developers of the PHQ-9.

Of the non-allegiant studies, 13 (42%) did not include a conflict of interest statement.[30–32 37 42 46 48 53 55 60 62–64] Similar to the algorithm studies, the newer studies were more likely to include a conflict of interest statement. Funding was acknowledged by most studies (27/31) and most received funding from academic and/or health research institutions. One study[57] acknowledged that the last author involved in the development of one of the instruments (CORE-OM), 'but does not gain financially from its use'. One study[51] acknowledged funding from industry, AHA Pharmaceuticals Roundtable, but stated that 'the funding organisations had no role in the design or conduct of the study, collection, management, analysis or interpretation of data; or preparation, review or approval of the manuscript. Fine *et al.* disclosed that the last author had financial and consulting interests (Pfizer was not cited as one of them).[56]

### Diagnostic test accuracy

Pooled sensitivity of allegiant studies was 0.87 (95% CI 0.77 to 0.93), pooled specificity was 0.87 (95% CI 0.76 to 0.94) and the pooled DOR was 49.31 (95% CI 25.74 to 94.48)—see table 5. Heterogeneity was moderate ($I^2$=55.1%). Figure 2 represents the sROCs for this group.

Pooled sensitivity of non-allegiant studies was 0.76 (95% CI 0.67 to 0.83), pooled specificity was 0.88 (95% CI 0.85 to 0.91) and the pooled DOR was 24.96 (95% CI 14.81 to 42.08), approximately half that of the allegiant studies (table 2). Heterogeneity was high at $I^2$=81.5%. Figure 2 represents the sROCs for this group.

The meta-regression for the studies using a cut-off point of 10 or above with allegiance status of the predictor showed that allegiance status was a significant predictor of the DOR (p=0.015) and explained 19.0% of observed heterogeneity.

### Quality assessment

The results of the quality assessment using the QUADAS-2 are given in table 4. For the patient selection domain, the two groups of studies were broadly comparable on two items (consecutive or random sample, avoid case-control design). However, all allegiant studies were rated as avoiding inappropriate exclusions (5/5) in contrast to 58% (15/26) of the non-allegiant studies.

On the index test domain, there were a number of differences between the two groups of studies. More of the non-allegiant studies (81%, 21/26) reported that the PHQ-9 was interpreted blind to the reference test compared with 60% (3/5) of the allegiant studies. All

**Table 5** Pooled estimates of diagnostic properties of the Patient Health Questionnaire-9 at cut-off point 10 and using algorithm scoring method in the non-independent vs independent studies groups

| Settings | No. of studies | No. of patients | Sensitivity (95% CI) | Specificity (95% CI) | Pooled positive likelihood ratio (95% CI) | Pooled negative likelihood ratio (95% CI) | Diagnostic OR (95% CI) | Heterogeneity: $I^2$ |
|---|---|---|---|---|---|---|---|---|
| Manea *et al*, 2014 SR–RA group | 7 | 4065 | 0.77 (0.70 to 0.84) | 0.94 (0.90 to 0.97) | 14.97 (8.39 to 26.71) | 0.23 (0.17 to 0.31) | 64.40 (34.15 to 121.43) | 78.9% |
| Manea *et al*, 2014 SR Independent studies | 21 | 9900 | 0.48 (0.41 to 0.91) | 0.94 (0.91 to 0.95) | 8.26 (6.15 to 11.09) | 0.54 (0.48 to 0.62) | 15.05 (11.03 to 20.52) | 68.1% |
| Moriarty *et al*, 2015 SR–RA group | 5 | 6188 | 0.87 (0.77 to 0.93) | 0.87 (0.76 to 0.94) | 7.24 (3.74 to 14.03) | 0.14 (0.08 to 0.25) | 49.31 (25.74 to 94.48) | 55.1% |
| Moriarty *et al*, 2015 SR Independent studies | 26 | 13 164 | 0.76 (0.67 to 0.83) | 0.88 (0.85 to 0.91) | 6.72 (5.06 to 8.92) | 0.26 (0.19 to 0.37) | 24.96 (14.81 to 42.08) | 81.5% |

SR, Systematic review; RA, researcher allegiance.

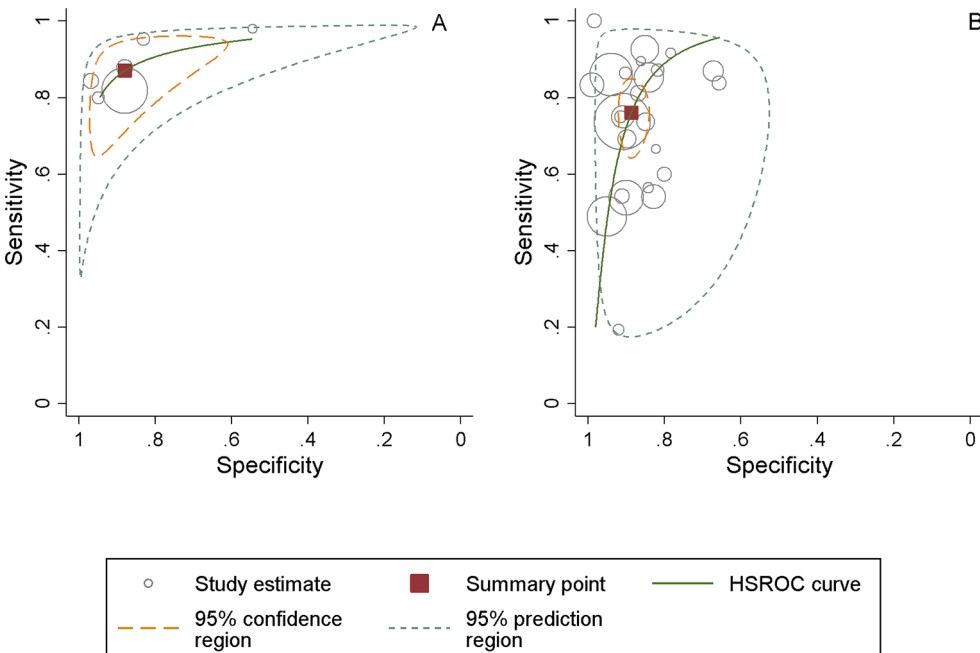

**Figure 2** Patient Health Questionnaire-9 summed items scoring method at cut-off point 10 summary receiver operating characteristic plot for diagnosis of major depressive disorder in allegiant studies (panel A) and non-allegiant studies (panel B). Pooled sensitivity and specificity using a bivariate meta-analysis. HSROC, hierarchical receiver operating characteristic.

(5/5) allegiant studies were rated as prespecifying the threshold on the PHQ-9 compared with 73% (19/26) of the non-allegiant studies. The two sets of studies were broadly comparable in terms of two items from the reference test domain (correctly classify target condition, reference test interpreted blind). Only one allegiant study used a translated version of the index test or reference test, so it is not possible to comment on differences between the two sets of studies in terms of these items from the index or reference test domains. For the flow and timing domain, the two groups of studies were broadly comparable for two of the criteria (interval of 2 weeks or less, all participants receive same reference test). However, fewer than half of the non-allegiant studies met the criterion for 'all participants included in the analysis' (42%, 11/26), whereas all allegiant studies met this criterion.

## DISCUSSION

This is to our knowledge the first systematic examination of a possible 'allegiance' or authorship effect in the validation of screening or case-finding psychological instrument for a common mental health disorder. We reviewed diagnostic validation studies of the PHQ-9, a widely used depression screening instrument. We found that allegiant studies reported higher sensitivity paired with similar specificity compared with non-allegiant studies. When entered as a covariate in meta-regression analyses, allegiance status was predictive of variation in the DOR for both the algorithm scoring method and the summed-item scoring method at a cut-off point of 10 or above.

Previous research has proposed several possible explanations for the allegiance effect.[9–11] One possibility is the

advertent bias that may serve to inflate the performance of a test when evaluated by those who have developed it. However, before concluding that the differences are due to this, it is important to explore and rule out alternative explanations. First, it is possible that any observed differences are a result of differences in study characteristics of the two sets of studies (eg, setting, clinical population). Second, differences in the methodological quality of the studies may also account for any differences. These possibilities are examined below.

### Difference in study characteristics as potential alternative explanations

The two sets of studies were broadly comparable in terms of gender and the prevalence of depression, so these variables are unlikely to offer an explanation for the differences. While there were some indications from both sets of comparisons that the PHQ-9 may have been researcher-administered more often in the independent studies, it is not immediately clear how this would lead to lowered diagnostic performance.

The diagnostic meta-analyses of the PHQ-9[13 14] have shown that the sensitivity and DOR of the PHQ-9 tends to be lower in hospital settings for both algorithm and summed-item scoring methods. While the fact that proportionally more non-allegiant algorithm studies were conducted in secondary care could explain the lower sensitivity and DOR values in the algorithm studies, in the studies that reported the cut-off point of or above this would not be the case as proportionally more allegiant studies were conducted in hospital settings.

Similarly, differences in the proportions of studies using translated versions of the PHQ-9 are also unlikely to offer

an obvious explanation of the difference in diagnostic performance, because in the algorithm set of studies more of the allegiant studies used a translated version of the test, but the proportions were in the opposite direction for the studies using a cut-off of 10 or above. We tested this by carrying out a sensitivity analysis restricting the sample to English studies and studies with adequate translation. The allegiance effect was still predictive of DOR variation between allegiance and non-allegiance studies variation in both algorithm (p=0.00) and summed item scoring at cut-off point of 10 meta-analyses (p=0.02).

A similar conclusion is also likely to apply to the age of the samples. There were more older adults studies in the non-allegiant than allegiant studies in the algorithm comparison. Depression could be more difficult to identify in older adults due to physical comorbidities that may present with similar symptomatology to depression and could account for the lower diagnostic performance in the non-allegiant studies. However, the non-allegiant samples in the studies that reported the psychometric properties at cut-off point 10 or above had younger samples than the allegiant studies, so this would not support this interpretation.

The SCID was used as the gold standard in nearly all allegiant studies. The fact that some non-allegiant studies used other gold standards could potentially explain the poorer psychometric properties of the PHQ-9 in these studies. The SCID is often regarded as the most valid of the available semi-structured interviews used in depression diagnostic validity studies as the reference standard. If we assume that this is the case and, furthermore, that the PHQ-9 is an accurate method of screening for depression, then the PHQ-9 may be more likely to agree with the SCID than other reference standards. However, when we carried out a sensitivity analysis restricting the sample to SCID-only studies, the allegiance effect was still predictive of DOR variation between allegiance and non-allegiance studies variation in both algorithm (p=0.01) and summed item scoring at cut-off point of 10 reviews (p=0.02).

### Differences in methodological quality as potential alternative explanations
The quality of the studies was evaluated using the QUADAS-2. Although there were several potential methodological differences between the two groups of studies from the algorithm papers, not all of these offer obvious explanations of the observed differences and some are unlikely as explanations. For example, more allegiant studies ensured that the reference test was interpreted blind to the index test. This is unlikely to account for the observed differences, because a lack of blinding is typically associated with artificially increased diagnostic performance, which is in the opposite direction to the pattern of results observed here. The impact of some other differences is less clear-cut. For example, a higher number of the non-allegiant studies met the criterion for consecutive referrals. For this to provide an explanation of the observed differences, the non-consecutive nature

of the referrals in the studies by those who had developed the PHQ-9 would need to have led to the overinclusion of true positives or underinclusion of false negatives given that these studies tended to report higher sensitivity relative to the non-allegiant studies (and vice versa for the independent studies). It is not immediately obvious how this would occur. The allegiant studies were more likely to have met the criterion of 'included all participants in the analysis'. It is possible that the greater loss of participants from the non-allegiant studies may have artificially reduced the observed diagnostic accuracy, although, again, it is not immediately obvious how this would have affected the true positive and false negative rates. Although there is not an obvious explanation of how these differences in methodological quality could account for the observed differences in diagnostic performance, it is important to recognise that they cannot on that basis be ruled out.

There are, however, two differences in methodological quality among the algorithm studies that are clearer potential alternative explanations. The higher rate of appropriate translations among the allegiant studies is potentially important, because lower diagnostic estimates may be expected from studies that have poorly translated versions of the index test. In the flow and timing domain, more allegiant studies ensured that there was a less than 2-week interval between the index and reference test. This is consistent with lower diagnostic performance in the non-allegiant studies: as the interval increases it is likely that depression status may change and this would lead to lower levels of agreement between the index test and the reference test.

There were also differences on some quality assessment items between the two sets of studies in the summed item scoring method comparison. The threshold was reported as prespecified in all allegiant studies in contrast to approximately three-quarters of the non-allegiant studies. On the face of it, this is unlikely to explain the observed differences, because the use of a prespecified cut-off point is likely to be associated with lower not higher diagnostic test performance. One possibility, however, is that studies that performed poorly at this cut-off point were less likely to be reported by those who had developed the measure. As discussed in more detail in the 'Limitations' section, we were unable to explore this possibility through the use of formal tests for publication bias.

All allegiant studies avoided inappropriate exclusions compared with approximately half of the non-allegiant studies. While this is a potential alternative explanation of the differences, it is not immediately obvious how this would explain the differences in diagnostic performance between the two sets of studies. Fewer than half of the non-allegiant studies met the criterion for 'all participants included in the analysis', in contrast to all of the allegiant studies met this criterion, but again this difference should usually work against the inclusive studies, not those excluding cases. More of the non-allegiant studies reported that the PHQ-9 was interpreted blind to the

reference test. This does offer a potential explanation, because the absence of blinding may artificially inflate diagnostic accuracy.

## LIMITATIONS

The results of this review need to be viewed in light of the limitations of the primary studies that contributed to the review and the review itself. An important consideration is to establish whether any observed differences between the diagnostic performance of the non-allegiant and allegiant studies are better accounted for by study characteristic or methodological differences. Caution, however, is needed in interpreting any differences, because of the small number of allegiant studies in both the algorithm and cut-off 10 or above comparisons. The small number of allegiant studies also meant that we were also unable to explore the potential role of publication bias in the non-allegiant and allegiant studies. At least 10 studies are required to use standard methods of examining publication bias, but the number of allegiant studies in both the algorithm and cut-off 10 or above comparisons were fewer than this. Papers published from August 2013 onwards are not covered in the literature search used and so it potentially misses some more recent studies that would be eligible for inclusion, although it is unlikely that many, if any, new allegiant studies have been published since.

## CONCLUSIONS AND IMPLICATIONS FOR FURTHER RESEARCH

The aims of the review was to investigate whether an allegiance effect is found that leads to an increased diagnostic performance in diagnostic validation studies that were conducted by teams connected to the original developers of the PHQ-9. Our analyses showed that diagnostic studies conducted by independent/ non-allegiant researchers had lower sensitivity paired with similar specificity compared with studies that were classified as allegiant. This conclusion held for both the algorithm and cut-off 10 or above studies. We explored a range of possible alternative explanations for the observed allegiance effect including both differences in study characteristics and study quality. A number of potential differences were found, although for some of these it is not clear how they would necessarily account for the observed differences. However, there were a number of differences that offered potential alternative explanations unconnected to allegiance effects. In the algorithm studies, the studies rated as allegiant were also more likely to use an appropriate translation of the PHQ-9 and were also more likely to ensure that the index and reference test were conducted within 2 weeks of each other, both of which may be associated with an improvement in observed diagnostic performance of an instrument. The majority of studies in both meta-analyses did not provide clear statements about potential conflict of interest and/or funding; however, the newer studies were more likely to provide such statements, which may reflect increasing transparency in this area of research.

We cannot, therefore, conclude that allegiance effects are present in studies examining the diagnostic performance of the PHQ-9; but nor can we rule them out. Conflicts of interest are an important area of investigation in medical and behavioural research, particularly due to concerns about trial results being influenced by industry sponsorship. Future diagnostic validity in this area should as a matter of routine present clear statements about potential conflicts of interest and funding, particularly relating to the development of the instrument under evaluation. Future meta-analyses of diagnostic validation studies of psychological measures should routinely evaluate the impact of researcher allegiance in the primary studies examined in the meta-analysis.

**Acknowledgements** One of the authors of this paper (SG) was supported by the NIHR Collaboration for Leadership in Applied Health Research and Care Yorkshire and Humber (NIHR CLAHRC YH). The views and opinions expressed are those of the author(s), and not necessarily those of the NHS, the NIHR or the Department of Health.

**Contributors** LM led on all stages of the review and is the guarantor. We used an established database of diagnostic validation studies of the PHQ-9 (Manea *et al.*, 2015; Moriarty *et al.*, 2015). SG provided expert advice on methodology and approaches to assessment of the evidence base. AM carried out the literature searches, screened the studies, extracted data and assessed the quality of the included studies for one of the systematic reviews (Moriarty *et al.*, 2015). LM carried out the literature searches, screened the studies, extracted data and assessed the quality of the included studies for the other systematic review (Manea *et al.*, 2015), analysed the data for both systematic reviews and drafted the report. JB involved in the development of the study, wrote the introduction section of the review and contributed to the production of the final report. DM supervised the quality assessment, methodology and approaches to evidence synthesis, provided senior advice and supported throughout and contributed to the production of the final report. All parties were involved in drafting and/or commenting on the report.

**Funding** LM was an NIHR Clinical Lecturer when this research was carried out. The NIHR had no role in the study design, methods, data collection, analysis or interpretation of data, nor any role in the preparation of the manuscript or decision to submit the manuscript for publication.

**Competing interests** None declared.

**Provenance and peer review** Not commissioned; externally peer reviewed.

**Data sharing statement** No additional data are available.

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
