## [Reviewer comments · BMJ Open]

ARTICLE DETAILS

TITLE (PROVISIONAL)	Are there researcher allegiance effects in diagnostic validation studies of the PHQ-9? A systematic review and meta-analysis
AUTHORS	Manea, Laura (proxy) (contact); Boehnke, Jan Rasmus; Gilbody, Simon; Moriarty, Andrew; McMillan, Dean

VERSION 1 - REVIEW

REVIEWER	Dr. Simon Hatcher The University of Ottawa Canada
REVIEW RETURNED	22-Dec-2016

GENERAL COMMENTS	I think it is well written and clearly discusses the research question it set out to answer. I think it would help readers and users of the PHQ-9 if the authors also included pool positive and negative predictive values using the pooled prevalence of depression in the study samples, (as well as sensitivity and specificity), as this is what really matters to clinicians. I also think it would be helpful for readers if the authors could be clear that the cut off score for the PHQ-9 is a score of 10 or above. At present the authors state the cut-off score is 10 which is somewhat ambiguous (a score of 10 could be above or below the cut-off). There is only one typo on line 166 where the first "of" should be an "if".
---

REVIEWER	Evangelos Evangelou University of Ioannina Medical School, Greece
REVIEW RETURNED	06-Jan-2017

GENERAL COMMENTS	The authors have performed a systematic search to investigate the allegiance effect in diagnostic validation studies. It has been found in other fields that the allegiance effect can act as a non-financial conflict of interest or optimism bias and tends to provide inflated or even exaggerated results in the presence of allegiance. Major comment: Allegiant researchers should include their status as a non-financial conflict of interest but this is not always the case. It seems that the authors have captured this information, therefore a sensitivity analysis on the articles that did not include this statement in the manuscript or in the acknowledgements section would be of interest information could be informative, even though I understand
--

	that this analysis would be based on a couple of studies only Minor comments; -Line 176, the sentence is broken -I would suggest using the terms allegiant and non-allegiant study instead of non-independent and independent study -The authors report a heterogeneity estimate but it is unclear where this refers at (I presume it is DOR?). Also, I understand that the bivariate modelling takes into account the heterogeneity when jointly synthesises the sensitivity and specificity to give summary estimates (sensitivity, specificity, DOR etc) and an I2 is not provided as in conventional methods of data synthesis
--	--

REVIEWER	Robert D. Gibbons University of Chicago United States
REVIEW RETURNED	06-Feb-2017

GENERAL COMMENTS	The authors should further explore some of the potential moderators of the observed differences in sensitivity (true positive) either through sensitivity (robustness) analyses that restrict the sample (e.g. SCID only) and/or using their random-effects meta-regression model to examine the SCID by independence interaction. While the number of studies is small, they could pool the studies with the different diagnostic thresholding methods (algorithm vs PHQ9>10, since they are similar in terms of the magnitude of the independence effect) and estimate the SCID effect (i.e. interaction). A similar approach could be used to examine other key potential confounders like translation.
--

VERSION 1 – AUTHOR RESPONSE

Reviewer 1

No Comment/Suggestion Response/Changes

1. I think it is well written and clearly discusses the research question it set out to answer. We thank the reviewer for these comments.
2. I think it would help readers and users of the PHQ-9 if the authors also included pool positive and negative predictive values using the pooled prevalence of depression in the study samples, (as well as sensitivity and specificity), as this is what really matters to clinicians. Thank you for this comment and we appreciate the point about the clinician's perspective (three of the authors are active clinicians themselves). But as mentioned in the request, predictive values are dependent on prevalence. Because prevalence is variable it has been questioned whether it is meaningful to combine predictive values across studies. Their aggregation in systematic reviews has therefore been discouraged. (Zwinderman & Bossuyt, 2008) Moreover, bivariate meta-analyses of sensitivity and specificity (as in our study) are widely accepted alternatives to bivariate meta-analyses of likelihood ratios and predictive values and do not face the additional challenge of pooling prevalences. For these methodological reasons and since our manuscript provides all data to quickly calculate predictive values for any specific context a reader might be interested in, we did not add this additional layer of analyses. (Leeflang, Deeks, Rutjes, Reitsma, & Bossuyt, 2012)
3. I also think it would be helpful for readers if the authors could be clear that the cut off score for the PHQ-9 is a score of 10 or above. At present the authors state the cut-off score is 10 which is

somewhat ambiguous (a score of 10 could be above or below the cut-off). Thank you for pointing this out. We have corrected this in the manuscript.

4. There is only one typo on line 166 where the first "of" should be an "if". Thank you for highlighting this, we have corrected the typo.

Reviewer 2

No Comment/Suggestion Response/Changes

1. Allegiant researchers should include their status as a non-financial conflict of interest but this is not always the case. It seems that the authors have captured this information, therefore a sensitivity analysis on the articles that did not include this statement in the manuscript or in the acknowledgements section would be of interest information could be informative, even though I understand that this analysis would be based on a couple of studies only. We are grateful for this important point brought up by the reviewer, which underlines the intricacies of catching allegiance status. We went through the studies again, but as is presented in our paper (full Col statements for all studies in tables 1 and 2 and discussed in the manuscript, only one study was in our sample that was allegiant and had a Col statement. (Löwe et al., 2004) Therefore no further subgroup analyses could be conducted based on this criterion.

Therefore we opted to stick to our admittedly rather conservative definition of "allegiance", since, although it is criticisable as any definition of a criterion, it is based on information readily available to any reader of the original studies.

We further hope that the concern about reporting practices, which we share with the reviewer, is sufficiently addressed by our statement in the conclusion section (last page of the manuscript): "The majority of studies in both meta-analyses did not provide clear statements about potential conflict of interest and/or funding, however the newer studies were more likely to provide such statements, which may reflect increasing transparency in this area of research."

2. Line 176, the sentence is broken Thank you for pointing this out. We have separated it in 2 sentences.

3. I would suggest using the terms allegiant and non-allegiant study instead of non-independent and independent study We agree, thank you for suggesting this. We have changed the text accordingly.

4. The authors report a heterogeneity estimate but it is unclear where this refers at (I presume it is DOR?). Also, I understand that the bivariate modelling takes into account the heterogeneity when jointly synthesises the sensitivity and specificity to give summary estimates (sensitivity, specificity, DOR etc) and an I2 is not provided as in conventional methods of data synthesis Thank you for pointing this out. The following clarification was added in the manuscript .

Heterogeneity was assessed using I2 for the diagnostic odds ratio, an estimate of the proportion of study variability that is due to between-study variability rather than sampling error. We considered values of $\geq 50\%$ to indicate substantial heterogeneity.(University of York. NHS Centre for Reviews and Dissemination., 2009)

Reviewer 3

No Comment/Suggestion Response/Changes

1. The authors should further explore some of the potential moderators of the observed differences in sensitivity (true positive) either through sensitivity (robustness) analyses that restrict the sample (e.g. SCID only) and/or using their random-effects meta-regression model to examine the SCID by independence interaction.

While the number of studies is small, they could pool the studies with the different diagnostic thresholding methods (algorithm vs PHQ9 >10 , since they are similar in terms of the magnitude of the independence effect) and estimate the SCID effect (i.e. interaction).

A similar approach could be used to examine other key potential confounders like translation. Thank you for this suggestion. We have carried out a sensitivity analysis restricting the sample to SCID only studies in line with the reviewer's first suggestion. The allegiance effect was still predictive of DOR variation between allegiance and non-allegiance studies variation in both algorithm ($p = 0.01$) and summed item scoring at cut-off point of 10 meta-analyses ($p = 0.02$).

Thank you for this suggestion, which we added in the manuscript: We have carried out a sensitivity analysis restricting the sample to English studies and studies with adequate translation. The allegiance effect was still predictive of DOR variation between allegiance and non-allegiance studies variation in both algorithm ($p = 0.00$) and summed item scoring at cut-off point of 10 meta-analyses ($p = 0.02$).

We refrained from analysing further potential moderators in detail, since some of them are discussed in previous publications (Manea et al., 2015) (Moriarty et al., 2015); and the main focus of this paper was on a theory-driven re-analysis of the set of studies, which required an in-depth qualitative discussion of our findings. We are happy to supplement further analyses if the reviewer sees glaring omissions of tests of alternative explanations with view to our research question and findings.

VERSION 2 – REVIEW

REVIEWER	Evangelos Evangelou University of Ioannina Medical School, Ioannina, Greece
REVIEW RETURNED	15-May-2017

GENERAL COMMENTS	Authors have adequately responded to my queries.
--

REVIEWER	Robert Gibbons University of Chicago United States
REVIEW RETURNED	25-May-2017

GENERAL COMMENTS	The authors have been responsive to the review. The final version should include more details of the analysis (e.g. sensitivity and specificity) of the new sensitivity analyses (e.g. language and SCID only studies).
---

VERSION 2 – AUTHOR RESPONSE

Reviewer: 2

Reviewer Name: Evangelos Evangelou

Institution and Country: University of Ioannina Medical School, Ioannina, Greece

Competing Interests: None declared

Authors have adequately responded to my queries.

Reviewer: 3

Reviewer Name: Robert Gibbons

Institution and Country: University of Chicago, United States

Competing Interests: None declared

The authors have been responsive to the review. The final version should include more details of the analysis (e.g. sensitivity and specificity) of the new sensitivity analyses (e.g. language and SCID only studies).

Authors' response:

We could extend the appendix by describing the two analyses in more detail. The authors did not feel this should be part of the main paper.